# Towards Sustainable Urbanization. Learning from What's Out There

**Alys Solly** * , **Erblin Berisha** and **Giancarlo Cotella**

Interuniversity Department of Regional, Urban Studies and Planning (DIST), Politecnico di Torino,
Viale Mattioli 39, 10125 Turin, Italy; erblin.berisha@polito.it (E.B.); giancarlo.cotella@polito.it (G.C.)
* Correspondence: alys.solly@polito.it

**Abstract:** The incremental recognition of the importance of land as a finite resource has led to the adoption and implementation of an increasing number of sustainable land use practices in European cities and regions. This paper reflects on these experiences, building on the evidence collected in the framework of the ESPON SUPER pan-European research project. In particular, the authors look at the project's database, which includes 235 examples of sustainable urbanization interventions gathered from all around Europe. In doing so, they reflect on the outcomes of these interventions, focusing on both their scope and objectives and the types of instruments that were adopted in their implementation. The objective of this contribution is to critically analyze the rich set of practices collected throughout the project and to provide guidance for decision and policy makers aiming at promoting a more sustainable use of land. In this light, it suggests a number of recommendations and warnings, bearing in mind that no "right instruments" or "right targets" exist that could prove successful for all European cities and regions.

**Keywords:** sustainable land use; urbanization; spatial governance and planning; Europe; ESPON

## 1. Introduction

Over the years, and especially since the Second World War, land transformation has become more and more intense, leading to the overexploitation of land and to the progressive recognition of its finite nature. More recently, the COVID-19 crisis has further warned us about the importance that a present and future sustainable built, as well as natural, environment, could have in facing unexpected emergencies more resiliently [1]. There is, therefore, an increasing need to find and adopt integrated solutions to make present and future development more sustainable [2]. Thus, it is essential for policy and decision makers to take careful decisions on urbanization and land use management, approaching the latter not only as a political and technocratic decision but as one that affects our society's well-being and quality of life [3].

This perspective is well acknowledged at the European level, with the European Union (EU) which, through time, has introduced a growing number of policies and actions aiming at promoting a more sustainable approach to development and urbanization [4]. In particular, the EU is trying to halt excessive land transformation with its objective to achieve zero net land take by 2050 [5] and, more recently, the European Green Deal has stressed the need to make Europe climate neutral by 2050 [6]. As a result, in the last few years, policy and decision makers at all territorial levels have started to dedicate increasing efforts to pursue urbanization and land use models that are more sustainable, thus leading to the consolidation of an increasingly heterogeneous set of interventions and practices aiming at this direction [7]. At the same time, it should be noted that this has happened from both the top-down and the bottom-up levels, in the context of both urbanized and depopulated remote rural areas [8,9].

Research and studies on sustainable urbanization and land-use have also increased through time, often stemming from different definitions and interpretations of "sustain-

ability". When focusing on the use of land, the most recent definitions of sustainable urbanization perceive the latter as the "design of future urban development as well as the re-development of existing ones in an environmentally friendly and resource-efficient manner" [10] (p. 1). In particular, sustainable land use seems to depend both on the socio-economic processes that trigger spatial development and on the effectiveness of the instruments that regulate these processes [3]. Adopting a similar perspective, the recently concluded ESPON SUPER project (Sustainable Urbanization and land-use Practices in European Regions, 2018–2020; https://www.espon.eu/super (accessed on 15 March 2021)) reviewed the multiple approaches put in place in different European cities and regions towards the achievement of a more sustainable urbanization, bearing in mind that there are no "right instruments" or "right targets" that could possibly fit all territorial contexts, also due to the high heterogeneity that characterizes the European continent in terms of socioeconomic development, administrative culture and spatial governance and planning [11,12].

The present contribution builds on the results of the SUPER project to develop guidance for decision and policy-makers aiming at promoting a more sustainable urbanization of their territories. It does this through a critical analysis of the rich set of practices collected throughout the project and, in particular, exploring the variable degree of success that characterizes interventions aiming at different goals, as well as adopting different types of instruments. After this introduction, the second section introduces the theoretical framework of the SUPER project, before section three describes the methodology that it adopted to collect and analyze sustainable urbanization and land use practices throughout Europe. Section four constitutes the core of the paper; it provides a quantitative overview of the collected interventions, in particular in relation to their localization and degree of success, to then discuss more in depth their results in relation to their scope and objectives (i.e., densification, regeneration, containment, governance and sectoral policies), as well as to the types of instruments that they have employed (i.e., visions and strategies, rules and legal devices, land use regulations, programs and projects). Finally, a concluding section rounds off the contribution, summarizing its main messages and the implications for decision and policy makers and introducing a number of avenues for future research.

## 2. Theoretical Framework

Despite its rather long history, sustainability as a concept is still characterized by multiple interpretations and rather blurred boundaries. [13]. The term was coined by the International Union for Conservation of Nature and Natural Resources (IUCN) in the early 1980s, and then adopted by the Brundtland Report a few years later, to indicate "development that meets the needs of the present without compromising the ability of future generations to meet their own needs" [14] (p. 54). As a result, during the last three decades, sustainable development has been at the center of the international research agenda [15]. Indeed, it has increasingly taken a central position in recent EU regional and urban politics, as evidenced for example by the decision to undertake the ESPON SUPER project. Nevertheless, although there is a "vast array of available best practices, little is known about the ways in which best practice is constructed, used, and contested, or of its implications for urban sustainability" [16] (p. 1029). Drawing on the consolidated literature on the topic, the ESPON SUPER project understands sustainability as characterized by three main aspects: temporal, thematic and institutional balance (see Figure 1). In particular, the *temporal balance* refers to the capacity to maintain long-term sustainable development for future generations and to enable the satisfaction of their needs [17]. Although often underestimated, certain factors, such as a governance quality and the durability of policies (e.g., the stability of funding), seem to effectively enhance sustainable urbanization processes [18] (p. 2). Moreover, among the characteristics that support the effectiveness of urban regeneration processes are those policies that envisage long-term sustainable targets, and which are supported by a strong political will (e.g., the UK brownfield targets, see Section 4.1.2). On the other hand, the *thematic balance* refers to three dimensions generally

referred to as the "three Es" (i.e., economy, ecology, equity) [19] or the "three Ps" (i.e., people, planet, profit) of sustainability [20]. According to the literature, any intervention faces the challenge of reconciling and enabling the coexistence of these dimensions [21]. Consequently, each of these dimensions has to be fulfilled without sacrificing the others, in order to achieve a development trajectory that is truly sustainable [22]. Finally, sustainability also depends on the *institutional balance*. For various reasons, institutional sustainability is a challenging issue for those working towards the development and implementation of sustainable urbanization policies. In fact, decisions and actions should be implemented through transparent and effective mechanisms, in line and coherent with the overall institutional framework in which they are adopted [23]. Thus, sustainable development can also be seen as a "social and political construct and, as such, the study of the operationalization of sustainable development through the implementation of specific policies provides the critical focus for research" [24] (p. 1).

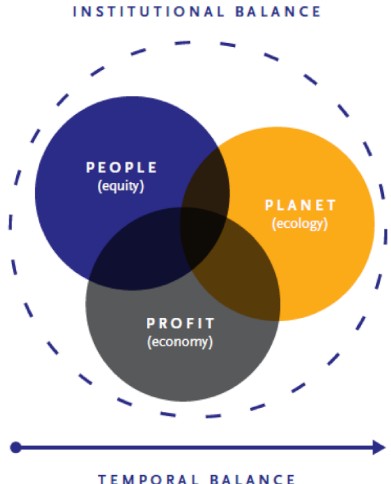

**Figure 1.** Understanding sustainability (source: [25] (p. 11)).

When it comes to understanding urbanization as a phenomenon that can occur more or less sustainably, the SUPER project does not merely refer to the movement of population to cities or the expansion of the built-up area, but to all physical developments that may affect land (homes, roads, construction sites, playgrounds, airports, business parks, etc.) and to the way they are continuously influenced by policies aiming at regulating and steering development and land-use [26]. In this light, the SUPER project did not measure urbanization in Europe only in quantitative terms but also had the ambition to conceptualize it as the outcome of the countless collective and individual decisions made by humans every day about where and how they want to live, work and play within the constraints of what they can afford and what they can access. In particular, whereas urbanization patterns can be quantitatively described on the basis of key drivers like demography, economic development and society/technology (e.g., [27]), the crucial decision to convert a site from a non-urban use to an urban use is governed by the payoffs and interests of the various actors involved, which, over time, can be described as development practices. Various drivers at the macro level, including institutional and policy drivers, create (dis)incentives at the micro level to create a "local regulatory regime" or "rules of the development game" [28]. Key agents with decision-making authority, those with legal rights or economic or political clout, then interact to produce a decision on land use.

To do this, the SUPER project has designed a conceptual framework that illustrates the main cause and effect relationships that influence urbanization and land-use change mechanisms (Figure 2). The left side represents the key drivers (e.g., demography, economic development, society and technology) of urbanization and land-use patterns. In this respect, the project aimed at highlighting the drivers of change (demand) that affect land use and the institutional aspects that affect urbanization (supply). The right side of the diagram

indicates the physical outcomes of land-use decisions (which then impact on the economy, society and environment) in the different European regions, which can be measured through the use of satellite imagery and monitored over time through quantitative datasets and qualitative evidence. In order to link the drivers of urbanization and land-use and their outcomes in European regions, the research gathered and analyzed multiple examples of land use interventions throughout the European countries, in so doing aiming at opening the black box of those local practices that actually contribute to shape land-use through time.

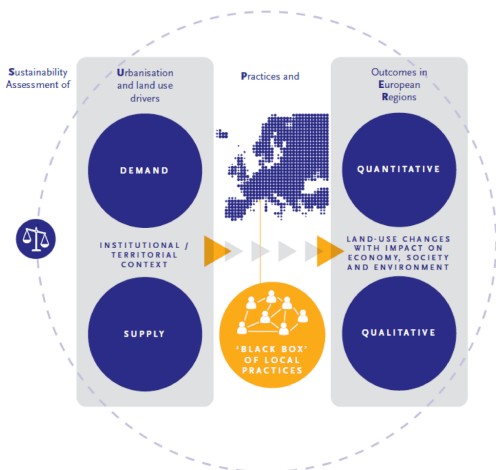

**Figure 2.** The Sustainable Urbanization and Land-Use Practices in European Regions (SUPER_ conceptual framework (source: [25] (p. 15).

In particular, the project acknowledges that the degree of success of any intervention aiming at steering urbanization in a specific direction is context-dependent (as certain forms of urbanization might be more sustainable than others in the different contexts) and that certain spatial governance and planning systems seem to be better equipped in relation to the promotion of sustainable urbanization than others [29]. In this light, it adopted a practice-oriented approach to explore how sustainable urbanization and land use is pursued in the different European countries and regions, thus developing a database of interventions that were explored in relation to their objectives and scopes, the types of instruments they employed, and their degree of success.

### 3. Methodology

*3.1. Data Gathering*

In order to collect sustainable urbanization and land-use practices from all around Europe, four main methodological steps were identified:

(i) first, a preliminary list of interventions was identified on the basis of the knowledge and experience of the SUPER consortium partners;

(ii) this list was complemented with examples retrieved from the national questionnaires of another ESPON applied research project [30];

(iii) then an online survey was created ad hoc, to reach out to experts from a number of pan-European organizations;

(iv) finally, the database was complemented and enriched through a thorough analysis of the scientific literature (e.g., articles, international research reports, national laws and regulations), in order to fill as much as possible the geographical and information gaps.

Throughout the project, the database underwent a number of quality control steps, performed by both the consortium partners and the Project Support Team, so it was subject to incremental fine-tuning. The online survey provided the highest number of results, generating over 160 responses. It was completed by national experts covering all the EU countries, as well as the EU candidate countries (i.e., Albania, North Macedonia,

Montenegro, Serbia and Turkey) and the remaining countries of the Western Balkans (i.e., Bosnia and Herzegovina and Kosovo). The experts were selected on the basis of a variety of different profiles: academic and scientific (e.g., universities, research centers) and more administrative roles (e.g., national, subnational, local agencies). The survey was then disseminated to the ESPON national contact points and members of the monitoring committee, as well as to the members of various academic and professional associations, such as the Association of the European Schools of Planning (AESOP), the European Council of Spatial Planners (ECTP-CEU) and the International Society of City and Regional Planners (ISOCARP). At a later stage, it was also circulated to experts through the use of social media (ResearchGate, as well as the AESOP and ESPON newsletters).

The survey focused on the current state of urbanization processes in the various countries. To facilitate the experts, it started with the following definition of sustainable urbanization and land use: "Sustainable land use means using and managing land assets in a way that does not compromise the livelihood of future generations. It implies a balanced consideration of social, economic, and environmental goods and services provided by the land uses in a certain region. It also implies a careful consideration of long-term attributes of resilience and robustness of the underlying ecosystem." [31] (p. 3). After that, it required the respondents to answer a short set of questions concerning the level of sustainability of urbanization and land-use in their country, the main impediments to the latter, and some examples of interventions affecting the sustainability of urbanization and land use in the practice (Table 1). Importantly, each expert was required to identify up to three interventions responsible for influencing the overall sustainability of urbanization and land use and, for each intervention, to point out the degree of success in terms of sustainable land use.

**Table 1.** Questions composing the SUPER online survey (source: author's own on the basis of [31] (p. 3)).

| | |
|---|---|
| 1. | In which country do you work? |
| 2. | In which sector do you (mainly) work? |
| 3. | We'd like to know if you think urbanization and land use in your country has become more or less sustainable (1 = much less sustainable, 5 = much more sustainable). Please explain why. |
| 4. | We want to learn about interventions (from territorial governance and spatial planning) that affect urbanization and land-use, for example policies, regulations, subsidies or strategies. These can be at the national or regional but also at the local level. The effects could be intentional or unintentional and could lead to sustainable or unsustainable outcomes. Could you provide some examples of these? Please include the name, the location, a short description and your assessment of its success (max. 3 examples). |
| 5. | What do you consider to be the most important impediment(s) to sustainable urbanization and land-use in your country? Please, briefly motivate your answer. Respondents could choose between: (i) lack of political will and/or declared policy aims in this direction; (ii) scarce effectiveness of the existing territorial governance and/or spatial planning instruments; (iii) other issues (e.g., corruption, lack of resources, lack of knowledge and data etc.). |
| 6. | Do you have any additional suggestion for our research team? (e.g., good sources or case studies to look into, or some additional insight from your region). |

### 3.2. Data Analysis and Intervention Assessment

After the data were collected, the intervention database was compiled and the collected interventions were further analyzed by reviewing available online documentation. The knowledge and the different skills of the consortium partners, as well as a careful analysis of the literature, helped to fill missing information and data. The interventions were then classified according to various categories and a number of analytical fields (see Table 2).

**Table 2.** Fields adopted in the analysis of the interventions (source: author's own elaboration on the basis of [31] (p. 4)).

| Categories | Fields |
|---|---|
| **Basic information** | <ul><li>Name of the intervention</li><li>Year (or time frame)</li><li>Country</li><li>Location</li><li>Scale (on the basis of NUTS classification)</li><li>Type(s) of EU territory involved (Urban, Rural, Functional area, Coastal area, Mountain region, Peripheral border, Cross-border, scarcely populated, Other)</li><li>Urban typology (if urban: Monocentric, Polycentric, Dispersed, Linear, Coastal)</li></ul> |
| **Characteristics** | <ul><li>Scope and objectives (Densification, Regeneration; Containment, Governance, Sectoral priorities (transport, environment, rural development)</li><li>Type of instrument (Visions and strategies, Rules and legal devices, Land use regulations, Programs, Projects)</li><li>Status (Statutory and mandatory, Statutory and non-mandatory, Non-statutory)</li><li>Level of coercion (Non-binding, Self-binding, binding for public actors, Binding for all actors)</li><li>Intervention inspired by the EU (Yes/No)</li></ul> |
| **Effects** | <ul><li>Description (scope, goals and functioning)</li><li>Description (in terms of how it works)</li><li>Degree of success—according to the goal of the intervention</li><li>Degree of success with respect to sustainable urbanization</li><li>Temporal sustainability: does the intervention prevent economic, social or environmental costs from being passed on to future generations?</li><li>Thematic sustainability: does the intervention advance values in the economic, social or environmental dimension without sacrificing those in other dimensions?</li><li>Institutional sustainability: is the intervention financially and politically sustainable over time?</li><li>Implementation quality—with respect to traditional evaluation criteria (is the intervention efficient-extent to which resources are well-spent, effective-extent to which goals were achieved, and relevant—for identified needs and problems?).</li></ul> |

Particularly relevant in the context of this paper is the categorization in relation to the scope and objectives of the interventions and the instruments that they adopted in order to achieve these objectives. More in detail, according to their scope and objectives, the collected interventions were subdivided as aiming either at promoting *densification* (e.g., up-zoning, infill development), fostering the *regeneration* of unused and/or problematic sites (e.g., land redevelopment, urban renewal), the *containment* of urbanization processes (e.g., green belts, urban growth boundaries), the introduction of specific *governance* models and mechanisms (e.g., cross-sectoral integration, integrated plans) or the achievement of specific *sectoral policies* (e.g., related to transport, environment or rural development). At the same time, the collected interventions were also subdivided in relation to the different types of instrument that were employed in each case in order to achieve the identified objectives, as for instance through the joint development of *visions and strategies* (e.g., strategic plans, guidance documents, etc.), *rules and legal devices* (e.g., national and regional laws), *land use regulations* (e.g., zoning, local plans), *programs* (e.g., economic incentives and other types of funds) and *projects* (e.g., single spatial transformation actions and initiatives). As shown in Table 2, the success of the interventions gathered through the above steps was then assessed in relation to both their explicit goals as well as to their ability to come to terms with the different dimensions that characterize sustainable urbanization and land-use—i.e., temporal, thematic and institutional sustainability. This assessment has contributed to develop an understanding of the factors that determine the success of an intervention or the possible reasons behind its failure, in so doing providing interesting evidence upon which to develop guidance for decision and policymakers aiming at promoting sustainable urbanization and land-use.

## 4. Results and Discussion

Through the described methodology, it was possible to collect and analyze as many as 235 practical examples of how, in the various European countries, actors active at the different territorial levels try to achieve a more sustainable urbanization and land-use (Figure 3). First of all, it is important to highlight that the collected sample is representative but certainly not exhaustive in describing the ongoing urbanization processes and the interventions put in place to address and steer the latter throughout Europe. In particular, whereas the database includes interventions from as many as 39 European countries, the number of interventions collected for the various countries is rather uneven and, whereas this may depend on the differential attention devoted in the various contexts to the issues at stake, it is also influenced by the localization of the consortium partners and of the respondents—with Germany, Italy and the Netherlands featuring a higher number of interventions. For the countries that were less represented in the sample, specific members of the ESPON Monitoring Committee and Contact Point were contacted multiple times in order to indicate additional potential respondents, who were then contacted and engaged, thus contributing to partially rebalancing the database. Overall, even though certain countries are better represented than others, the collected interventions provide a rich and rather comprehensive overview of the recent efforts put in place to promote sustainable urbanization in Europe.

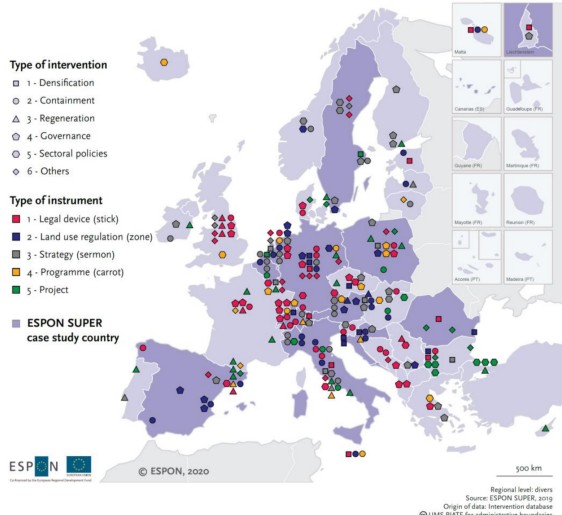

**Figure 3.** The localization of the SUPER intervention throughout Europe (source: [25] (p. 35)).

As mentioned, each intervention was qualitatively assessed in relation to its success in promoting a more sustainable use of land. This assessment phase was based on the responses of the country experts to the online survey and further verified through desk research (e.g., analysis of scientific articles and reports retrieved through the web). By crossing the level of success with the other variables, it was possible to reflect on which goals and which types of instrument have been more successful, and to further explore the reason behind the success [3] (p. 4). In particular, as shown in Table 3, for each category of the chosen variable, the interventions show varying degrees of success in relation to their ability to achieve a more sustainable urbanization.

**Table 3.** Degree of success of the analyzed interventions in relation to their scope and goals and to the type of instrument they adopted (source: authors' own elaboration).

| Type | | Degree of Success | | | | |
|---|---|---|---|---|---|---|
| | | 1 | 2 | 3 | 4 | 5 |
| Scope and goals | Densification | 9% | 0% | 41% | 36% | 14% |
| | Regeneration | 3% | 7% | 28% | 28% | 34% |
| | Containment | 7% | 7% | 32% | 34% | 20% |
| | Governance | 5% | 2% | 59% | 25% | 9% |
| | Sectoral policies | 12% | 0% | 28% | 40% | 20% |
| Type of instrument | Visions and strategies | 4% | 2% | 40% | 38% | 17% |
| | Rules and legal devices | 11% | 5% | 42% | 29% | 14% |
| | Land use regulations | 10% | 3% | 44% | 31% | 13% |
| | Programs | 4% | 13% | 13% | 35% | 35% |
| | Projects | 21% | 6% | 26% | 21% | 26% |

The degree of success is related to the interventions' ability to promote sustainable urbanization and land-use (1 = unsuccessful; 2 = scarcely successful; 3 = mixed success, 4 = almost successful; 5 = successful).

As regards their scope and goals, the interventions that promote regeneration seem to be the most successful: in fact, the majority of interventions (62%) are graded four and five, and 28% are graded three. On the contrary, interventions that promote containment seem often scarcely successful: 14% of the interventions are graded one and two. As far as the types of instruments are concerned, programs seem to be the most effective: a high number of interventions (70%) are graded four and five, and 13% are graded three. On the contrary, projects often produce outcomes which are scarcely successful: in fact, 26% of the interventions are graded one and two.

Starting from these first results, the next section carries out an in-depth exploration of the causes that might influence the level of success of the interventions that adopt: (i) different scopes and objectives, and (ii) different types of instruments. The results provide interesting reflections and insights for all those stakeholders who are appointed to take decisions and who are involved in the technical drafting of policies aiming at a more sustainable urbanization.

### 4.1. Scope and Objectives of the Analysed Interventions

This subsection looks at a number of interventions that seem to promote sustainable land use, as well as limiting land take effects. The interventions are presented in relation to their scope and objective (densification, regeneration, containment, governance and sectoral policies) and, for each of them, the effectiveness and the level of success is discussed. As pointed out in the methodology section, the majority of the examples are based on the experts' responses to the online survey. In some cases, additional sources were investigated in order to verify the information and gather further knowledge.

#### 4.1.1. Densification

In general, densification strategies seem to encourage different typologies of urban development (e.g., compact, polycentric). The results of the intervention database show that densification types of interventions, which promote up-zoning and in-fill development mechanisms, are usually successful in encouraging more sustainable urbanization processes.

Certain characteristics, such as the adoption of long-term sustainable aims and objectives, seem to support the effectiveness of these types of strategies. This can be seen in the Croatian Spatial Plan of Primorje-Gorski Kotar County (available at: https://zavod.pgz.hr/en/plans_and_reports (accessed on 15 March 2021)), which aims to limit future urban growth by promoting a more effective management of land use. To limit urban growth, a number of criteria for determining the size of building areas of settlements, regulating population density, were set. For example, the maximum surface area of building areas in each municipality was derived from the projected population and

the minimum density of the inhabitants of the urban area. However, the intervention also seems to have had negative effects since, in an attempt to limit future urban growth, non-residential facilities seem to have been driven further away. Thus, it is important for the local administrations to adopt tangible and short-term objectives when trying to promote a long-term sustainable development.

Other characteristics, such as an increased cooperation between the various stakeholders, also seem to improve the effectiveness of these interventions. For example, the success of the Royal Seaport eco-district in Stockholm (Sweden) is attributable to the constant dialogue and negotiation between the various actors (e.g., public and private) during the various phases of the project. The project shows how the City of Stockholm (which has limited space for greenfield development) has managed to promote densification measures in order to be able to accommodate population growth, as well as to find the most effective environmental solutions [32].

The implementation of legal binding instruments often seems to improve the success-fulness of these types of interventions. This is the case of the general development plan of the City of Stara Zagora and its adjacent territories (Bulgaria). For the expert reporting this intervention in the survey, it is one of the most effective tools to fulfil its limitation functions. In fact, the objectives and targets of the plan, including the upper limits of the development indicators, are compulsory and the failure to comply with them is illegal.

Data, knowledge and technical capacity are other characteristics that seem to promote more effective sustainable development. For example, the Infrastructural Cost Calculator, a strategy set up in the region of Lower Austria (Austria), supports municipalities in pre-assessing the financial costs (e.g., municipal infrastructural costs and tax revenues) of urban expansion and related population increase [33]. Thus, the strategy tries to assess the municipal consequences on where and how new inhabitants are settled. However, the effectiveness of the intervention seems to vary in relation to its implementation and a certain level of discretionality seems to characterize its implementation.

### 4.1.2. Regeneration

Urban regeneration is another goal which frequently concerns interventions aiming to promote sustainable urbanization and land use. A number of characteristics, such as those that envisage the concept of reuse, support the effectiveness of this type of intervention. This can be seen in many urban regeneration processes, such as: Gründachstadt Linz (Austria), the transformation of green roofs [34]; Réinventer Paris (France), the renovation of underutilized areas; Dublin Docklands (Ireland), the regeneration of brownfield areas [35]; the transformation and renewal of vacant areas in Berlin (Germany)unused since World War II.

Among the characteristics that promote urban regeneration are also those that envisage long-term sustainable targets. For example, since 1998 the United Kingdom has been applying brownfield targets (with at least 60% of new housing to be built on brownfield land by 2008), under the banner of an "urban renaissance" [36].

Improved multilevel cooperation between stakeholders also seems to strengthen the effectiveness of these types of interventions [37]. In Italy, the community-led regeneration process in Casoria produced very positive results in relation to the rehabilitation of abandoned areas and the enhancement of public participation. As stated by the expert reporting this intervention, the vision for the regeneration of a former sulfur mine (the Solfatara) in Manziana, through a collaborative and inclusive stakeholder participation in the context of common land ownership and management, is another interesting Italian initiative. On the contrary, the regeneration and rehabilitation of parts of the Taht-el-Kale Quarter in the City of Nicosia (Cyprus) was perceived as less successful. The initiative was part of a wider sustainable integrated urban regeneration strategy and worked in synergy with various social and cultural projects already implemented in the area. Nevertheless, the intervention was considered as less effective, according to the expert, mainly due to the scarce level of public participation.

The adoption of legally binding instruments often improves the successfulness of regeneration interventions. This is the case of the 2007 zero-growth plan of Cassinetta di Lugagnano (Italy) [38], which prohibits urban expansion in order to keep agricultural land as intact as possible. To do this it facilitates the repurposing of existing buildings and the regeneration of industrial areas. Finally, the adoption of an integrated approach also seems to help interventions to be more successful. This can be seen in the 22@Barcelona (Spain) regeneration program [39], which was well-integrated with the restructuring process of the metropolitan area and the urban policies framework.

It is important to highlight that speculation mechanisms can worsen the level of success of these interventions. Indeed, in contrast to the above cases of successful regeneration processes, some of Istanbul's housing renewal projects (Turkey) were criticized by the expert reporting this intervention for being driven by speculation, leading to high-rise housing in peripheral areas, without social infrastructure and transport facilities.

### 4.1.3. Containment

In the last decades, many containment interventions have been implemented in Europe with the objective of reducing land take. As a consequence, a number of sustainable strategies and green belts have been designed (e.g., the Grüner Ring in Leipzig, Corona Verde in Torino) to limit and control urban growth. Numerous interventions of this kind have proven successful in promoting sustainable development. For example, the Corona Verde strategy [40] envisages an ecological 'crown' around the metropolitan area of Torino (Italy), and brings together different intersectoral policies in order to reduce urban land consumption and to increase the quality of the rural–urban environment (e.g., through the mitigation and renaturation of infrastructural barriers, the conservation of the rural heritage).

Certain characteristics, such as the support of a strong political will and the adoption of long-term visions, seem to improve the implementation of these interventions. For example, the German government set the 30 hectares' target, with the ambitious goal of reducing annual land consumption to 30 hectares per day nationwide by 2020 (Umweltbundesamt—UBA, German Environment Agency: www.umweltbundesamt.de/en/ (accessed on 15 March 2021)). Cooperation that goes beyond municipal boundaries is another characteristic that often improves the successfulness of containment interventions. This is the case of Vision Rheintal (Austria) [41]. For the expert informant, its success is partly due to intra-municipal cooperation, as well as engagement with a heterogeneous group of experts (thus, promoting the transfer of expert knowledge) and the adoption of a holistic approach.

The adoption of legal binding instruments also seems to improve these interventions. For example, the 2014 Tuscany Regional Law on soil consumption (Italy) requires municipalities to delimit the borders of their more densely urbanized areas and to promote the urbanization of empty plots through simplified regulations and incentives. Non-residential transformations outside urbanized areas, which involve the consumption of new land, are only allowed if the co-planning conference provides a favorable opinion (Legge Regionale Toscana 65/2014). Similarly, the 2009 Law for the City of Sofia (Bulgaria), which works together with the city's General Urban Development Plan (GUDP), is considered to have produced positive outcomes, in particular by stating that "the designation of existing green plots or parts thereof in the urbanized territories, created according to the development plans cannot be changed" (art. 9). The GUDP, however, seems to have been less successful. In fact, inconsistencies seem to exist between the plan's overall goals and some of its measures and implementation tools [42]. Thus, certain interventions, if not implemented correctly, might lead to a discrepancy between the desired objectives and the actual outcomes. This might also be exacerbated by a lack of political will, technical capability and scarcity of economic resources.

Moreover, certain containment initiatives may turn out to be counterproductive for the promotion of sustainable land use. This seems to be the case of the Cork Area Strategic Plan (Ireland), which provides a long-term vision for the development of the Cork City-Region

up to 2020. The expert reporting the intervention noted that, even though it aims to reduce urbanization in the countryside, an overexploitation of natural resources still occurs, and that the strategy is based on a pro-growth approach.

### 4.1.4. Governance

Governance interventions that try to improve the mechanisms through which governmental stakeholders manage urban and rural areas seem to influence the ways sustainable development is carried out at regional and local levels. However, these types of interventions seem to have produced results that are more varied than those presented above.

Certain characteristics, such as when interventions promote a long-term sustainable development perspective and adopt an integrated approach, are usually more effective. For example, in Stockholm (Sweden), the urban transformations and modalities of integrated planning are considered successful cases of integrated land use, housing and transport planning. Nevertheless, multi-level collaboration in Stockholm's urban transformations favoring the integration of local actors has had to face challenges, such as the intervention of the central government [43]. In Helsinki (Finland), the agreements on land use, housing, and transport (MAL) for the 2016–2019 period are also perceived as successful. In fact, the intervention promotes a more effective land use management and cooperation between municipalities.

As regards the adoption and implementation of urban plans, governance interventions seem to have had diverse impacts in the different cities and regions. In general, multilevel collaboration seems to improve the effectiveness of these types of interventions. In Poland, the 2016 planning law and housing policy of the Warsaw metropolitan area is a positive intervention, which has contributed to improving the spatial structure of both the city and its surrounding area, in the light of long-term sustainable development (e.g., green corridors, protecting green areas, reducing sprawl). Likewise, the Tri-City metropolitan area planning (Poland) aims to promote a harmonious development of the coastal area of Gdansk-Gdynia-Sopot, enhancing public transport. The intervention is generally perceived as successful due to the integrated governance structure it set up; however, despite its good potential, more time is still needed to fully assess its success. On the contrary, the attempt to promote bottom-up, integrated metropolitan planning led to the approval of the Poznań metropolitan area planning law (Poland) that, despite identifying areas that are important for environmental protection (e.g., degraded areas that require urgent revitalization actions), failed to achieve the expected results in terms of municipal coordination.

### 4.1.5. Sectoral Policies

Sectoral policies that refer to transport (e.g., mobility), environment (e.g., air, soil, water) and rural development (e.g., agriculture) seem to have different impacts on sustainable land use. Overall, as can be seen in the interventions presented in this subsection, it seems that the adoption of a more integrated policy approach, as well as a long-term strategy or vision, leads to a more sustainable development.

As regards transport policies, the Urban Mobility Plan of Barcelona (Spain), introduced "the superblock model" [44,45], an intervention that is considered very successful since it reduced air pollution levels. In the United Kingdom, the Mini-Holland in Waltham Forest (www.walthamforest.gov.uk/content/creating-mini-holland-waltham-forest (accessed on 15 March 2021)) is another successful intervention that supports urban mobility, reducing motorized transport and creating segregated cycle lanes on the model of Dutch-style infrastructure. The results of the Slovenian Sustainable Urban Mobility Plans (SUMPs) seem to be less successful, according to the expert informant, even though the country adopted the "EU Sustainable mobility for a prosperous future" strategy in order to manage urban mobility more effectively. In fact, only one third of the municipalities adopted the SUMPs and their poor acceptance by local political leaders seems to be one of the main challenges. Since SUMPs are not an obligatory instrument, providing financial support seems to be the best way to encourage their implementation. Another intervention whose

success is open to question is the City of Sofia's underground metro (Bulgaria) that seems unable to integrate its mobility goals with achieving a more integrated land use approach. The Lyon–Torino high-speed railway and tunnel project (between France and Italy) is also considered a less successful intervention due to the constant delays and conflicts it has generated [46,47]. In fact, the project has been contested by environmental associations over its potential impacts on the environment (e.g., consumption of land, exploitation of natural resources).

As regards environmental and rural development policies, in Germany, the expert informant considers the BOKS—Soil Protection Concept as a positive example of sectoral intervention, which promotes a higher level of environmental quality and aims to reduce soil consumption. Another interesting intervention is the Lower Austrian spatial planning ordinance for wind energy utilization, which sets up a framework to manage wind-park development up to 2030. It identifies areas where wind turbines are allowed and where development is severely restricted. The expert informant deems it a positive intervention because it promotes the safeguarding of the natural environment; however, the construction of wind turbines in green areas (e.g., in forestry areas) seems to be a controversial topic in the country. It is also worth mentioning the 2007–2013 Green cross-border area—investment in nature project (between Bulgaria and Serbia) which has enhanced environmental awareness, as well as an exchange of knowledge and good practices. On the contrary, in Austria, for the expert informant, the Soil Enhancement Plan has the potential to support sustainable urbanization and land use (e.g., it tries to retain high-quality soil), but is rarely applied. The flood management system along the Tisza River in Hungary is also considered by the expert informant as an unsuccessful intervention due to a lack of coordination between the authorities and financial mechanisms.

*4.2. Adopted Instruments*

Experience has shown there is no ideal tool to be used for managing land use. On the contrary, sustainable urbanization and land use could be achieved through the implementation of a variety of instruments. Examples are discussed below in relation to visions and strategies, rules and legal devices, land use regulations, programs and projects.

4.2.1. Visions and Strategies

Visions and strategies are future-oriented and non-mandatory instruments that set out the main directions for development. One of the characteristics of successful visions and strategies is establishing ambitious, future-oriented objectives and, even more importantly, identifying realistic ones; while conversely, underfunded, incoherent or unrealistic strategies can erode credibility and commitment [25]. On the basis of the examples gathered, strategies introducing an ambitious target that have influenced the use of land include the Vision Rheintal of Vorarlberg (2004, update in 2017) in Austria and the Tri-City metropolitan area planning (2007) in Poland. The objective of the former is to create an interconnected polycentric region, promoting cooperation within it, supporting cross-border cooperation and creating an interconnected living space, fostering and enhancing regional awareness and regional identity, while the objective of the latter is to have a harmonious, complete and dynamic development of the metropolis of Tri-City (Gdańsk, Sopot and Gdynia). Both initiatives promote a more integrated approach to urban containment by facilitating investment on e-mobility transportation, encouraging densification along public transport routes and improving intercity connections within the region. Another successful case is the Corona Verde in Italy where 81 municipalities banded together to promote a new and alternative vision of the territory based on the quality of the environment and high quality of life [40]. The success of the strategy is demonstrated by its capacity to mobilize substantial funds for implementing short-term projects within a wider long-term strategy. Another interesting strategy is the Kooperationsplattform Stadtregion (2014) of Salzburg in Austria. For the expert informant the strategy recognizes the negative impact of diffuse urbanization on quality of life and that is why it aims to limit fragmented settlement and

commercial development in the suburban belt of the main cities. To promote containment, the strategy implemented a regional green belt approach using development compensation measures to guarantee equal benefits for participants. At the national level, one clearly successful strategy is the zero-growth goal for car traffic (2018) applied in Norway that aims to introduce non-motorized models of transport [48].

However, visions and strategies are not always successful and face various challenges in addressing sustainable land use. This has proved the case for a number of strategies for European cities, which were challenged by sustainability trade-offs, implementation difficulties and lack of institutional will and capability. For example, the new Finger Plan of Copenhagen (2016–2019) to promote a more efficient transport network paved the way for sacrificing valuable green areas in the countryside [49]. Similarly, the Cork Area Strategic Plan in Ireland (2001–2020) aimed to reduce the loss of agricultural land, but in actual fact rural land consumption increased. Again, while the Athens Master Plan of 2014 introduced innovative concepts, it failed to combine its attention to environmental causes due to a lack of public consultation processes [50,51], while the Sustainable Metropolitan Plan of Rome Capital City 2003 has never been implemented due to limited political and institutional will [52]. Similarly, at the central level, the Climate Adaptation Program in Portugal shows that the success of this type of intervention can be undermined by a lack of political will at the local level [53].

### 4.2.2. Rules and Legal Devices

Sustainable land use can be addressed by establishing specific legal devices, such as binding laws and bylaws, to create a supportive institutional framework. Decision and policy makers can activate a plethora of different legal devices that can be mandatory or non mandatory—allowing authorities a certain level of flexibility. Sustainable land use can be promoted by introducing ad hoc laws and norms (for land use or environmental protection), as well as by promoting disincentive measures (fees, ad hoc taxes). Based on the experiences gathered, legal devices are not always successful. Contradictions emerge, for instance, in the case of the Poznan Metropolitan Area Planning Law (Poland) [54], which, despite having the merit of introducing concepts like "compact city" and "energy-efficient spatial structure", does not offer enough legal clarity to enforce them.

Sustainable land use can also be achieved by introducing successful economic disincentives or compensation mechanisms. Thus, various initiatives to disincentive excessive land use consumption have been widely experimented in Europe. Among others, it is worth mentioning the cases of the Development and Maintenance Fee applied in the region of Upper Austria (Austria), the double urbanization fee in Emilia Romagna (Italy) and the soil compensation account introduced in Dresden (Germany) [55]. In the Austrian case the initiative establishes that the infrastructure fee is the responsibility of the owner, in order to limit urban expansion, while the Emilia Romagna region decided (by resolution No. 186/2018), on the one hand, to double urbanization fees for projects that convert agricultural land into built up area and, on the other hand, to decrease these by at least 35% (local administrations are allowed to reduce them to 100% if necessary) for projects aiming at regenerating abandoned areas. Finally, the soil compensation account of Dresden aims to confine built-up land for settlements and traffic to 40% of the total urban land, as well as to force investors to carry out compensation measures by themselves or to pay a compensation fee.

An example of a successful land use initiative taken in Europe is the referendum to limit land take (2013) in Switzerland. The aim of the referendum was to curb urban sprawl and promote internal development, forcing municipalities to limit urban expansion. In fact, additional land can only be zoned if there is a real need for it [25]. This kind of direct democracy instrument is typically used for enhancing citizens' awareness on the issue and obtaining political legitimation for sensitive issues like land consumption. Even though not easily replicable—due to institutional mechanisms and cultural attitudes—the importance

should be taken into consideration of responsibilizing citizens towards land use, which can be done at a central, as well as at a local level, by increasing participatory mechanisms.

Another restrictive example of land use from Switzerland is the Weber Law (2012). This initiative is interesting from two different perspectives. Firstly, it aims to fight land consumption by limiting the construction of second homes to preserve Switzerland's natural landscape from overbuilding by pursuing containment objectives. Secondly, it establishes measurable targets: no more than 20% of a municipality's housing can be second homes otherwise there will be building restrictions. This is particularly useful for preserving touristic destinations from being overexploited and thus reducing the diffusion of empty or temporarily occupied building structures.

### 4.2.3. Land Use Regulations

Land use regulations establish binding principles, usually through zoning, that define how land can or cannot be transformed. Historically, this occurs through dedicated local land-use planning tools, aiming at regulating physical development or, in some cases, to forbid development and to leave the land as it is [56]. Based on the experiences gathered, plans are shown to act in different directions according to their final objective. Some plans may promote policies aiming at reducing land exploitation or increasing its optimal use (e.g., Municipal Operative Plans of Reggio Emilia and Bassa Romagna). In both cases, the decision was taken to reduce the buildable surface by 30% and 50%, respectively, to guarantee a more sustainable use of land, while preventing landowners from paying higher taxes on buildable land.

In relation to the overestimation of buildable areas, the municipal operative plan of the City of Reggio Emilia (Italy) was employed to reduce the number of areas which had been zoned for urban uses but remained unbuilt. Since landowners pay taxes based on the value of the zoned land, stripping development rights also yields a financial benefit. The cooperation between municipalities and landowners has succeeded in downzoning over 135 ha of potential urban land to rural functions since 2015. A second phase has so far removed an additional 70 ha from potential urbanization. In so doing, the municipality takes back the possibility of (re)organizing its territory without having any restriction or impediment to changing its planning trajectory. Similarly, the Province of Utrecht (Netherlands) is experimenting with the de-zoning of urban areas back to agricultural use via the imposed land-use plan, primarily regarding unbuilt office space [25].

Land use regulation can also contribute to reducing spatial competition, which has been recognized as one of the main drivers of diffuse urbanization among municipalities. In this respect, the Municipal Structural Plan of the Union of Municipalities (2009) of Bassa Romagna in Italy offers a good example of what can be done to limit competition among municipalities [57]. Based on a cooperative approach and the predisposition of an appropriate institutional arrangement, nine municipalities decided to come together in drafting planning tools to better address sustainable land use. The adopting of the new plan and the further consolidation of the "Union" as a level of administration have contributed to limit the potential negative impact of the divergent interests, through the introduction of a system of compensation across municipalities.

Other land use plans, instead, may focus mainly on protecting and improving existing agricultural land like the Territorial Action Plan of the Huerta de Valencia (2018) in Spain and the Rural Park South (1990) in Milan (Italy), or limiting urban expansion as done by the Physical Environment Special Plan Protection (1980) of the Andalucía Region in Spain.

However, land use regulations cannot guarantee per se the achievement of sustainable land use objectives. In some cases, plans can increase land transformation to respond to market mechanisms (see the Sofia General Urban Development Plan of 2007 in Bulgaria and the Spatial Plan of Zone Chalupkova of 2009 in Bratislava, Slovakia) [42]. Land-use regulations can also promote, indirectly, the explosion of informal development due to their rigidity or lack of clear implementation mechanisms. The Urban Development Plans (starting from 1999) of Prishtina in Kosovo, are an example that, despite their original

intentions, led to urbanization processes outside formal rules [58]. Similarly, even if the aim of the Outside Development Zones of 2006 in Malta is to safeguard the integrity of rural areas, they have been accused of justifying speculative initiatives as construction limits are easily exceeded.

### 4.2.4. Programs

Programs are policy packages aiming at a particular objective. They can be used to create economic conditions (financial schemes, direct investments, allocation of developing funds) for sustainable land use. Throughout Europe, these initiatives have been mainly implemented to create the economic conditions for the renewal of industrial areas (e.g., 22@Barcelona implemented in Spain since 2000), the protection of environmental quality (e.g., the Re-creation of Lake Karla in Thessaly in Greece since 1999 and the Enjoy Waltham Forest program of 2014 in the UK), as well as for promoting cross-cutting initiatives like the BENE—Berlin Programme on Sustainable Development (Germany), implemented since 2015, or the National Strategy for Inner Area (SNAI—Italy) [59,60]. More in detail, the Re-creation of Lake Karla in Thessaly [61,62] was seen as an opportunity to enhance the water supply, restore the ecosystem and improve the quality of the soil that was in danger of overexploitation. The Enjoy Waltham Forest program has also been positively seen because it has delivered a series of micro-interventions (e.g., segregated cycle lanes, planting trees) aiming at promoting a more environmentally oriented approach. More oriented towards spatial and social regeneration, the Piano Periferie 1 and 2, introduced in Italy since 2015, aim to recover abandoned and deprived areas by investing in environmental and social, as well as economic sustainability, by allocating 4 billion EUR (two have been already activated) for the improvement of the cities' peripheries by prioritizing urban requalification and the regeneration of abandoned areas. In this respect, several initiatives have been financed and some of them are already implemented, while others are expected to be concluded in the coming years. Finally, the success of the Berlin Program on Sustainable Development (BENE), is evidenced by the amount of funds allocated (234 million EUR), the number of projects put in place and the integration of existing development programs, and a similar assessment concerns the Italian SNAI, aiming at integrating the use of EU and domestic resources in rural areas.

### 4.2.5. Projects

Projects are individual ad hoc initiatives with a given timeframe, which can be used for the implementation of permanent or provisional transformations of sites. They are extremely heterogeneous in terms of nature, objectives, design and level of success. Various examples show how projects can contribute to regenerate abandoned areas, like the Dublin Docklands (Ireland) which started in 1997, the South Harbour in Copenhagen (Denmark) started in 1995 and the Royal Seaport in Stockholm (Sweden) started in 2008. The same has been done in other parts of Europe, like the Vila d'Este in Vila Nova de Gaia (Portugal) works concluded in 2015, the Industrial Park Borská Pole in the City of Plzeň (Czech Republic) in 1992 and the Miasteczko Wilanów initiative implemented in Warsaw (Poland) since 2002. Although diverse in some aspects, all the projects deal with recovering, eco-designing and promoting a healthy life-style. Efforts at reducing the human footprint have been made in the case of the Eco-Viikki project in Helsinki (Finland) implemented between 1999 and 2010, which demonstrates how new living standards can be successfully combined with a minimal impact on the environment. Similarly successful was the Caserne de Bonne in Grenoble (2003–2009), the first eco-district in France. From the sustainable land-use perspective, the crucial factor is that the shapes of the buildings were compact to reduce land consumption and urban sprawl. More community-oriented, but also successful, are the transformation of Vacant Urban Areas (1996) in Berlin (Germany) into attractive parks and vibrant public spaces [63], and the case of Rotterdam (Netherlands), where houses in deprived neighborhoods (since 2014) were simply bought up by the municipality and given

away for free to anyone willing to invest a certain amount in renovation and promising to live there for at least five years [64].

However, projects can also fail or create unexpected or unwanted effects. Regeneration initiatives can produce gentrification like the Urban Development Project of Hyllie developed between 2007 and 2013 in Malmö (Sweden) that ended up with an image of housing "wealthy white westerners" [65]. If not well-designed, regeneration projects may channel a pro-market authoritarian approach, as the cases of Skopje 2014 (North Macedonia) and the Belgrade Waterfront of 2015 in Serbia demonstrate. While both pursue the rehabilitation of strategic urban areas, local community interests take a back seat vis-à-vis private investors. Finally, some projects explicitly provide for overexploitation of natural resources like the Nessebar and Sunny Beach seaside development in Bulgaria since 1958, the Ranca Resort implemented since 1990 in Romania and the third Istanbul Bosphorus Bridge Canal Project in Istanbul in Turkey (2013–2016) [66].

## 5. Concluding Remarks

On the basis of the analysis of the various interventions collected in the framework of the ESPON SUPER project, it is possible to develop a tentative set of recommendations and warnings for decision and policy makers aiming at promoting a more sustainable urbanization of their territories. This concluding section rounds off the contribution by presenting these recommendations and warnings, with particular reference to the variable degree of success that actors may achieve when putting in place interventions aiming at different goals and adopting different types of instruments.

In particular, when looking at the scope and the different objectives of the analyzed interventions, the presented evidence shows that land use can be addressed in different ways, none of which, however, are either fully sustainable or unsustainable (Table 4). *Densification* can potentially contribute to achieving sustainable land use if opportunely addressed. For instance, interventions aiming at it have the potential to promote further social equity by reducing car dependency and journey distances [67]. According to the project's results, successful factors of densification are, among others, the adoption of a long-term perspective (e.g., up zoning and measures for infill development), as well as the introduction of legally binding instruments. As shown in the literature [68], densification does not always imply sustainable land use. In some cases, it may contribute to increasing traffic congestion if not opportunely designed [69]; in others, it has been shown to increase housing prices, while it also contributes to reducing green public areas in favor of buildings [70].

Similarly, some of the analyzed examples show that sustainable land use has been successfully promoted by regenerating abandoned areas. *Regeneration* of brownfields requires a paradigmatic shift that makes operative the concepts of reuse and of integrated sustainable development, thereby facilitating a circular use of land [71]. Even regeneration, however, cannot be taken for granted, as it may become more expensive than transforming greenfield sites [72] and thus not economically attractive for market operators due to the cost–benefit logic that development may give rise to. Another inhibitive factor of promoting regeneration is the fact that it may be intended as a tabula rasa without considering local specificity (and community needs) [73], thus paving the way for gentrification phenomena that can lead to social exclusion [74]. *Containment* oriented interventions are among the most common approaches in addressing sustainable land use. For instance, containment can be promoted by restricting the development of city edges, introducing policies to better contain urban expansion [75] and, in so doing, preserving agricultural land from being converted [76]. The information gathered by SUPER seems to suggest that one of the key factors in the success of containment initiatives is the presence of effective political will, since the spatial effect of these initiatives usually takes time to be visible. They also require the establishment of an effective and efficient normative apparatus (e.g., legally binding instruments) that can limit speculative market mechanisms (i.e., increased land prices, exclusion of certain social categories, concentration of development benefits, etc.). However,

containment also brings side effects when it comes to sustainable development, such as traffic congestion and an increase in housing prices. In particular, unclear containment strategies may pave the way to increasing land and houses prices, thus forcing individuals and businesses to relocate to areas where more space is available. This kind of spatial competition may reduce the development pressure in one area but drastically increase it in others, making it inconvenient (or undesirable) in terms of sustainable land use [77].

**Table 4.** Successful factors and pitfalls when it comes to promoting sustainable land use (authors' own elaboration).

| Scope and Objectives | Successful Factors | Pitfalls and Warnings |
|---|---|---|
| Densification | The adoption of a long-term perspective (e.g., up-zoning and measures for infill development). The inclusion and cooperation with private partners, as well as a balance between public and private interests. The adoption of legally binding instruments often improves the success of such interventions. | Densification may contribute to increasing traffic congestion if not opportunely designed. In some cases, densification has been shown to increase housing prices, which has a negative impact on affordable housing. Densification may contribute to reduction of green public areas in favor of buildings. |
| Regeneration | The adoption of a long-term vision (e.g., enhancing the economic, environmental and social quality of the area and of the local community). The application of the concept of reuse and of integrated sustainable development. Addressing environmental, economic and social issues at the same time. | Regeneration may become more expensive than transforming greenfield sites and thus not economically attractive for market operators. Regeneration may be intended as a tabula rasa without taking care of local specificity (and community needs). Regeneration may—in some cases—pave the way for gentrification and social exclusion. |
| Containment | Effective political will is needed since the spatial effect of containment initiatives usually takes time to be seen. The establishment of an effective and efficient normative apparatus (e.g., legally binding instruments) guarantees a certain level of success. The limitation of speculative market mechanisms (i.e., increased land price, exclusion of certain social categories, concentration of development benefits, etc.) | Unclear containment strategy may increase costs of land (and houses). If not carefully drafted, containment may force individuals and businesses to relocate to areas where more space is available (spatial competition). |
| Governance | Integrating public priorities with private (corporate or individual) interests. Establishing an adaptive multilevel collaboration and governance models: each context is different, as well as the contingencies where the political choices are taken. Implementation should be accompanied and supported by cooperative governance mechanisms able to include different scales (optimally both top-down and bottom-up approaches). | Uncoordinated governance models may act against sustainable development. Poorly defined responsibilities (or overlappings) are at the basis of uncontrolled development. |
| Sectoral policies | The adoption of an integrated approach and long-term sustainable perspective taking into consideration a multiplicity of sectoral interests. Stronger collaboration between the various stakeholders seems to be fundamental for achieving of a good level of sectoral integration and coordination. Support of soft initiatives that have direct and immediate impacts: long-term projects usually require more time to show their advantages. | The adoption of sectoral policies may lead to excessive policy fragmentation. Uncoordinated sectoral strategies may pave the way for unsustainable development. |

The mechanism of implementation and models of *governance* are also important in terms of addressing sustainable land use. The most successful ones seem those that integrate public priorities with private (corporate or individual) interests. Effective public and private partnership seems to limit eventual negative externalities that development initiatives may give rise to. According to the research's results, another important factor of successful governance is the establishment of adaptive multilevel collaboration, taking into account that each context is different, as well as the contingencies where the political choices are made. These multilevel governance relations should take care to achieve an optimum balance between top-down and bottom-up approaches. Conversely, uncoordinated governance models and the overlapping of responsibilities seem to act against sustainable development. Due to this complexity, land use can also be addressed by *sectoral initiatives*. As shown, there are a series of examples, throughout Europe, that illustrate how sectoral policies support sustainable land use. In this respect, the success of this kind of initiative may depend on the adoption of an integrated approach and long-term sustainable perspective taking into consideration a multiplicity of sectoral interests. A strong collaboration between the various stakeholders also seems to be fundamental for achieving a good level of sectoral integration and coordination. Accordingly, sectoral authorities should be further integrated in the planning process [78], since sectoral strategies have proven to impact on land use although their impact is not always positive.

The type of instrument to be adopted in order to pursue further sustainable urbanization trajectories is also a highly relevant factor and, in this case too, no one-size-fits-all solution seems available to policy makers (Table 5). For example, the use of *visions and strategies* have proven to be successful when they support common territorial perspectives for territories that share the same needs and challenges and activate cooperative decision-making mechanisms. They benefit, moreover, from the establishment of a strong, stable and future-oriented political will that in the long run makes the difference. On the other hand, visions and strategies may fail when the required leadership and/or institutional capacity to translate them into effective measures is missing or when the targets identified are too ambitious and not realistically implementable. That is why decision and policy makers should establish tailored targets in line with territorial needs and effective institutional readiness to translate them into practice. According to the information gathered, *rules and legal devices* have proven successful when they are clear in their final objective (e.g., limit land consumption, protect valuable natural areas, monitor the housing and rental markets) and normatively solid. This is particularly important considering their technical feasibility and the link with their social acceptability. On the contrary, these tools are often less effective when they envisage a large window of flexibility as a consequence of the possibility to interpret the norms discretionally. Failure may also depend on their capacity to address sustainable development holistically. In this respect, decision and policy makers should guarantee an acceptable equilibrium between the various sustainability dimensions (e.g., social, economic, and environmental). If visions and legal devices set the "rules of the game", *land use regulations* are often used to translate them into practices. Through the implementation of regulative plans, decision and policy makers have the opportunity to convert political will and technical capacity into effective land use transformation. That is why it is important to be aware of the factors that have been shown to successfully address sustainable land use. According to the sample, successful examples show an optimum balance between the need for development and the need to achieve sustainable land use. This can be obtained by reorienting planning decisions in order to promote sustainable land use, for instance by reconverting buildable areas into agriculture ones, or protecting land instead of allowing its exploitation. Conversely, planning tools are subject to failure if they directly legitimate speculative phenomena when it comes to facilitating private investments and real estate (gentrification, exclusion of disadvantaged social groups, etc.), while in certain cases they may indirectly facilitate illegal initiatives when plans are hard to implement (lack of effective implementation mechanism).

**Table 5.** Successful factors and pitfalls when it comes to promoting sustainable land use (authors' own elaboration).

| Type of Instruments | Successful Factors | Pitfalls and Warnings |
|---|---|---|
| Visions and strategies | Supporting common territorial perspectives for territories that share the same needs and challenges.<br>Decisions are based on cooperative mechanisms; otherwise, visions and strategies could remain on paper without any chance of being effectively implemented.<br>A strong, stable and future-oriented political will makes a difference. | There are no institutional capabilities to translate into effective measures.<br>Targets identified are too ambitious, wide in their content and not realistically implementable.<br>There is a lack of political perseverance. |
| Rules and legal devices | Should be clear in their final objective (limit land consumption, protect valuable natural areas, monitor the rental and housing markets).<br>Should be normatively strict and adapted to their different institutional contexts.<br>Should be technically feasible (coherent set of norms and regulations that may guarantee the applicability of interventions).<br>Should be socially acceptable (sustained by social legitimacy). | Legal devices are not strict but envisage some windows of flexibility (not mandatory).<br>Legal devices do not consider sustainability in a holistic perspective privileging one of its dimensions at the expense of the others. |
| Land use regulations | Optimum between the need for development and the need to achieve sustainable land use. Reorienting planning decisions in order to promote sustainable land use by reconfiguring (reconverting) buildable areas into agricultural ones.<br>They are used as instruments of land protection instead of land exploitation. These can be implemented by promoting measures of urbanization containment and protection of agricultural/natural land. | May address the various sustainability dimensions only to a partial extent. In particular, in many cases the environmental dimension appears more prominent than the economic and the social ones.<br>May give legitimacy to speculative phenomena when it comes to facilitating private and real estate investments;<br>May indirectly facilitate illegal initiatives when plans are hard to implement. |
| Programmes | Should be well-integrated with existing instruments and spatial planning tools and policies.<br>Should be operative-oriented by indifferently promoting mega-projects or small-scale initiatives.<br>Their design should integrate all the thematic dimensions of sustainability. | There is a gap between ambition and effective achievement possibilities (overestimation of economic capabilities).<br>They are too development-oriented instead of focusing on environmental protection.<br>They are not institutionally and economically well-coordinated with the rest of the programmes. |
| Projects | When they are part of a long-term territorial vision without, however, losing sight of short-term objectives.<br>When they incorporate simultaneously economic priorities (being cost-efficient), environmental needs (promoting pro-environmental solutions) and social aspects (supporting citizens' involvement). | Regeneration (and densification) sites are viewed as a tabula rasa for facilitating real-estate and speculative initiatives.<br>Projects are used for achieving political legitimacy or exercising political power.<br>Projects produce side effects like increasing inequalities, gentrification, segregation, etc.<br>Projects explicitly promote the overexploitation of natural resources since they follow pro-growth market logics. |

Moreover, the analyzed interventions have shown that even the most concrete strategy or plan may fail if it is not properly supported by effective *programming instruments*. These instruments have proven to be proactive in addressing land use, when well-integrated with existing spatial planning tools and policies. The capacity to mobilize funds effectively is one of the key factors of any initiative towards sustainable urbanization. Mobilizing funds also means implementing real land transformation by developing *projects*. Even though often underestimated, projects are the operative instruments that effectively transform

space. That is why it is important for decision and policy makers to know which are the successful factors that make projects work. In particular, projects are proactive in sustainable urbanization when they are part of a long-term territorial vision without, however, losing sight of short-term objectives. They help to make a strategy visible and concrete thereby creating social legitimacy. Yet, they may only prove fully successful when simultaneously incorporating economic priorities (being cost-efficient), environmental needs (promoting pro-environmental solutions) and social aspects (supporting citizens' involvement). At the same time, projects are often subject to market and political manipulation. Among other factors, policy and decision makers should be aware that in some cases the regeneration (and densification) of sites might facilitate speculative real-estate initiatives.

In conclusion, it is important to recall once again that urbanization processes are a combination of factors that cannot simply be replicated from one context to another, but require a tailored approach [25]. In this view, land use policies cannot be intended as pre-packaged, but should be contextualized according to territorial, institutional, and cultural specificities [79]. Sustainable land use is a polymorphic concept, whose approach can shift from a more ecological and environmental perspective by promoting reconversion of land, establishing ambitious target and strategies, thus promoting a densification of urban structures through the rehabilitation of industrial areas or applying a wide range of incentives and disincentives. In this light, there are a number of messages that decision and policy makers should take into account, namely: (i) to avoid "one size fits all" solutions and thus each policy recommendation should be assessed according to territorial specificities; (ii) to avoid stand-alone initiatives when addressing complex issues like sustainable land use (multi-dimensional, multi-sectoral and multi-stakeholder approaches are preferable); and (iii) to ensure that sustainable land use is a shared responsibility and the identified solutions should be carefully evaluated and shared with all the relevant actors. As pointed out in the Introduction, making careful and prudent decisions on land use is not only a political and technocratic decision but, as the COVID-19 pandemic has dramatically highlighted, also one with highly significant societal consequences [80].

In short, this paper emphasizes the current and future opportuneness of comparative land use studies in a world which is coming to terms with the crucial need to face increasingly challenging issues such as climate change and sustainability as pointed out in the UN Sustainable Development Goals [81]. Nevertheless, it is important to reiterate that, even though there is no "right instrument" or "right target" for all European regions, "right attitudes" exist that can be adopted to promote a more sustainable urbanization, and we hope that the present contribution may constitute a useful support in that direction.

**Author Contributions:** G.C. designed the project and supervised its implementation. A.S. and E.B. carried out data collection and analysis, under G.C.'s supervision. All sections of the paper were written and revised by all three authors. All authors have read and agreed to the published version of the manuscript.

**Funding:** The research was carried out as part of the ESPON SUPER 2019–2020 project (https://www.espon.eu/super (accessed on 15 March 2021)).

**Acknowledgments:** The research project SUPER (Sustainable Urbanization and Land-Use Practices in European Regions) has been carried out within the framework of the European Territorial Observatory Network (ESPON—https://www.espon.eu/ (accessed on 15 March 2021)). The consortium responsible for the project is coordinated by PBL (the Dutch Environmental Agency) and composed of Politecnico di Torino, BBSR (German Federal Institute for Research on Building, Urban Affairs, and Spatial Development), OIR (Austrian Institute for Regional Studies and Spatial Planning), University of Valencia, University of Warsaw and Urbanex. Additional information concerning the project is available at: https://www.espon.eu/super (accessed on 15 March 2021). The authors would like to thank all the members of the ESPON SUPER team for their proactive cooperation throughout the project.

**Conflicts of Interest:** The authors declare no conflict of interest.

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
