# Peer review of "Towards Sustainable Urbanization. Learning from What’s Out There"

_land, doi:10.3390/land10040356_

Round 1
Reviewer 1 Report
The paper present merits for publication. But need improve some sections mainly the theoretical section. The reference section is short only with 34 references. Need quoted more works.
Other related changes are two completely differentiated sections. 1. Theory and 2. methods.
In this reference Paniagua A (on line 2 june 2020) Geographies of differences (and resistances) in urbanized and depopulated remote rural areas. In Leal W et al (eds) Life and land. Encyclopedia of the U N Sustainable Development Goals. Springer Cham., authors can find abundant bibliographic clues
Author Response
Answer to reviewer 1
We would like to thank the first peer-reviewer for their helpful comments and suggestions and have revised the paper accordingly. Below, we present more in detail the way in which the various comments and suggestions have been taken into account.
Comments and Suggestions for Authors
The paper present merits for publication. But need improve some sections mainly the theoretical section. The reference section is short only with 34 references. Need quoted more works.
In this reference Paniagua A (on line 2 june 2020) Geographies of differences (and resistances) in urbanized and depopulated remote rural areas. In Leal W et al (eds) Life and land. Encyclopedia of the U N Sustainable Development Goals. Springer Cham., authors can find abundant bibliographic clues
The theoretical section has been improved and enriched, and new references have been added to provide corroboration and additional focus. In particular, the reference suggested by the reviewer has been used in the text and added to the reference list.
Other related changes are two completely differentiated sections. 1. Theory and 2. methods.
Following the reviewer’s suggestion, there are now two separate sections: 2. Theoretical framework and 3. Methodology.
Reviewer 2 Report
The article is very interesting, well written and fluent. However, I consider that the main aspect to improve is the presentation of the methodology, I do not consider that it is adequately explained, especially how the results that are presented are reached. Suggested modifications are detailed below.
The abstract requires a clear expression of the purpose of the article, which I consider is missing.
Regarding the objective, it is understood that it is the one expressed in line 58 and 59. However, the objective of the results (line 244 and 245) and methodology (line 126) do not match with it. I suggest reformulating the objective of the study, including all the contributions of this article. Or eventually include specific objectives for each section of the article.
Line 47: It is not clear who "they" refers to.
Line 50: It is preferable to put the acronym in parentheses and leave the full name in the text. Also, it could help fluency to put a footnote with the additional information, including the project link, and remove it from the main text.
In section two, there is a confusing subdivision of the topics. I consider it appropriate to clarify which is the general theoretical framework of the project and which is the effective methodology of this research. following some ideas to solve this confusion:
Line 74: It is suggested to add a subtitle that explains the content of this first part. It could be for example "definition of concepts used".
Title 2 could be renamed with "theoretical framework" and 2.1 as a third section named "methodology"
Line 76: Rephrase to remove repeats (sustain-sustainable)
Figure 2: This figure is not easy to understand. I consider it useful to explain more in the text how it should be interpreted.
Section 2.1. I believe that those expressed from line 127 are not methodological steps, but rather sources of information. The methodological steps that I understand are for example: information gathering, database elaboration, database analysis. I suggest dividing the methodology into methodological steps and dedicating a subchapter to each of the steps.
Furthermore, only the procedure carried out to prepare the survey is mentioned in detail, while the other steps for obtaining the results are not adequately explained, creating doubts.
In particular, it remains unclear how the practices that help the success of the interventions are determined, since they are neither in the fields of the database nor in the questions of the online survey.
Figure 3: From what I understand, the elements listed correspond to “Scope and Goals” and not to “type of intervention”. However, in the legend, it is reversed.
Line 232, 233: the legend can be included as a note to the table and excluded from the text.
Table 3: Specify which degree of success has been used to prepare this table (according to the goal of the intervention or according to sustainable urbanization).
It is difficult to interpret this table because the number of interventions for each type is different. An option could be to relativize all the data in percentage so that they can be more easily comparable.
The results present the practices that improve the effectiveness of the interventions. It is not very clear how these results are reached, and it is not clear by what criteria the projects mentioned as examples are chosen. Also, when a specific project is mentioned, it is not clear if it is the only one or if there are more to whom the same thing has happened, and if yes, how many.
Author Response
Answer to reviewer 2
We would like to thank the second peer-reviewer for their helpful comments and suggestions and have revised the paper accordingly. Below, we present more in detail the way in which the various comments and suggestions have been taken into account.
Comments and Suggestions for Authors
The article is very interesting, well written and fluent. However, I consider that the main aspect to improve is the presentation of the methodology, I do not consider that it is adequately explained, especially how the results that are presented are reached. Suggested modifications are detailed below.
The abstract requires a clear expression of the purpose of the article, which I consider is missing.
The abstract has been modified and the objective of the study has been clearly stated.
Regarding the objective, it is understood that it is the one expressed in line 58 and 59. However, the objective of the results (line 244 and 245) and methodology (line 126) do not match with it. I suggest reformulating the objective of the study, including all the contributions of this article. Or eventually include specific objectives for each section of the article.
The objective of the study is clearly stated in the abstract and in the introduction. In our opinion, it is now also coherent with the objectives of the various sections, all contributing to the overall goal of the article.
Line 47: It is not clear who "they" refers to.
The sentence has been modified.
Line 50: It is preferable to put the acronym in parentheses and leave the full name in the text. Also, it could help fluency to put a footnote with the additional information, including the project link, and remove it from the main text.
The position of the acronym has not been changed as it seems fine as it is. Since the project is presented (with the project link) within the main text a footnote does not seem to be necessary. Moreover, to our understanding the journal does not allow footnotes.
In section two, there is a confusing subdivision of the topics. I consider it appropriate to clarify which is the general theoretical framework of the project and which is the effective methodology of this research. following some ideas to solve this confusion:
Line 74: It is suggested to add a subtitle that explains the content of this first part. It could be for example "definition of concepts used".
Title 2 could be renamed with "theoretical framework" and 2.1 as a third section named "methodology"
Following the reviewer’s suggestion, there are now two separate sections: 2. Theoretical framework and 3. Methodology. Moreover, the methodology section has been further subdivided in order to help the reader grasp the various steps.
Line 76: Rephrase to remove repeats (sustain-sustainable)
The sentence has been modified following the suggestion.
Figure 2: This figure is not easy to understand. I consider it useful to explain more in the text how it should be interpreted.
A paragraph which describes Figure 2 more in detail has been added in the main text.
Section 2.1. I believe that those expressed from line 127 are not methodological steps, but rather sources of information. The methodological steps that I understand are for example: information gathering, database elaboration, database analysis. I suggest dividing the methodology into methodological steps and dedicating a subchapter to each of the steps.
The section has been revised and subdivided into three subsections.
Furthermore, only the procedure carried out to prepare the survey is mentioned in detail, while the other steps for obtaining the results are not adequately explained, creating doubts.
The procedure carried out to prepare the survey was the key step in terms of the results, which is why it is described in more detail. Reference is made to the SUPER project methodological report, that may provide further information about the other steps to the interested readers.
In particular, it remains unclear how the practices that help the success of the interventions are determined, since they are neither in the fields of the database nor in the questions of the online survey.
Table 2 shows in detail all the fields of information (comprising the degree of success with respect to the various aspects of sustainable urbanization) that were compiled for each intervention.
Figure 3: From what I understand, the elements listed correspond to “Scope and Goals” and not to “type of intervention”. However, in the legend, it is reversed.
The name “scope and goals” has been now adopted in all relevant sections of the paper.
Line 232, 233: the legend can be included as a note to the table and excluded from the text.
The legend has been added at the bottom of the table and excluded from the text.
Table 3: Specify which degree of success has been used to prepare this table (according to the goal of the intervention or according to sustainable urbanization).
The degree of success is now specified in the legend added at the bottom of Table 3, and also in the text.
It is difficult to interpret this table because the number of interventions for each type is different. An option could be to relativize all the data in percentage so that they can be more easily comparable.
The table has been modified and the data is now presented in percentages.
The results present the practices that improve the effectiveness of the interventions. It is not very clear how these results are reached, and it is not clear by what criteria the projects mentioned as examples are chosen. Also, when a specific project is mentioned, it is not clear if it is the only one or if there are more to whom the same thing has happened, and if yes, how many.
As explained in the methodology, more than 230 interventions were collected (mainly through the online survey) and analysed. After presenting the raw numbers in relation to the success of these interventions – subdivided by scope and objectives and types of instrument – for each section we present only some relevant examples, that were chosen as the most salient and explanatory of the overall situation.
Reviewer 3 Report
Review on manuscript:
Solly, A., Berisha, E. and Cotella, G. Towards sustainable urbanization. Learning from what’s out there. Land, 2020.
Summary
The manuscript summarizes a wide range of interventions and evidences of sustainable urbanization practices that the authors gathered through interviews, surveys and literature from European countries. The authors created a database with the information gathered and a scale to help reader evaluate the outcome of the interventions. The authors focus on the interventions scopes, outcomes and objectives and discuss the success or failure of the implemented approach in each case. Based on their results and the analysis and evaluation of the data and cases, they aim to develop guidance for policy makers. In that way the aim of this study is to be use as a useful guide in future designing of interventions considering urban land change.
Merits
The study focuses in sustainable land use and land change in urban areas. This is an important issue as urban areas are increasing in size and thus the way that these areas are designed is crucial, since they affect not only the environment but also our quality of life. The analyses that the authors use are appropriate for this study. The authors use various approaches to examine the interventions that take place in different European countries, their scopes and outcomes. The authors use online surveys and interviews and also examine the available literature. The literature cited is relevant to the study, but there are several instances, which have been noted, in which the authors don’t use appropriate literature to support their examples and cases. These assertions should be substantiating with references and there being noted in detail.
Critique
Introduction
The Introduction provides a good and general background on the main topic. That gives the reader a quick understanding of the main subject. The objectives of the study are clearly defined in the last paragraph.
Line 49; p.2. Citation [8] and citation [2] are referring to the same study. The authors need to check this and update the reference.
Theoretical Framework and Methodology
The technique used to obtain the results seems appropriate for this study, especially since the main focus is to demonstrate the different approaches of land use and urbanization design among the European countries.
It is not vital to the present study but something that could be interesting would be if the authors could demonstrate any differences in the evaluation of the interventions results obtained using the interview technique in contrast to the results of the literature.
Results and Discussion
The Results and Discussion Section is well constructed; divided in subsections using the Scope and goals and Type of Instrument factors as levels of categorization. The authors state in Section 2 that “the interventions were further analyzed by reviewing available online documentation”. However there are some instances that the authors did not substantiate their assertions with references and it is not clear to the reader when discussing their results whether a statement is supported from an interviewee’s survey or from literature.
Figure 3; p.7. The map is not very easy to read and symbols in neighbor countries in some instances are mixed up and not easy to distinguish between them (in example Belgium and Netherlands).
Table 3; p.7. This table presents absolute values. The numbers sum up the interventions that correspond to the degree of success (in a scale from 0 to 5) of the different types of scopes and instruments. But inside the study’s text the authors are using percentages to explain and present the results of this table. Thus is difficult to the reader to correlate the table with the text. The authors could consider adding percentages in the Table’s results.
Lines 264-266; p.9. The authors state that “this can be seen in the Croatian Spatial Plan...” This statement needs support with references.
Lines 311-310; p.9. The authors mention that the “regeneration process in Casoria produced very positive results...” and that “the rehabilitation of parts of the That-el-Kale Quarter in Nicosia... less successful”. This statement needs support with references.
The same as above applies to the “Case of the 2007 zero-growth plan of Cassinetta di Lugangano (lines 324-325; p.9) and in “the 22@Barcelona regeneration program” (lines 329-330; p.9).
The authors should also consider adding references to the examples of “the case of Vision Rheintal (Austria)” (lines 352-353; p.10) and “the 2014 Tuscany Regional Law” (lines 358-359; p.10).
References also must cited that describe the interventions of: “Mini-Holland in Waltham Forest” (lines 422-423, p.11), the “Slovenian Sustainable Urban Mobility Plans” (lines 424-425; p.10), the “City of Sofia’s Underground metro” (lines 431-432; p.10), the “Lyon-Torino high speed railway and tunnel project” (lines 432-435; p.11), the “BOKS-Soil Protection Concept” (lines 437-438; p.11) and the “Lower Austrian spatial planning ordinance” (lines 440-444; p.11), and the “project Green cross-border area”, the “Soil Enhancement Plant of Austria” and the “Tisza River in Hungary” (lines 445-451; p.12),
Intervention examples that need support with references are also: the “Tri City metropolitan area planning in Poland” (lines 465-466; p.12), the “Corona Verde case in Italy” (lines 474-478; p.12), the “Kooperationsplattform Stadregion in Austria” (lines 478-480; p.12), the “New Finger Plan of Copenhagen” (lines 490-491; p.12) and the “Strategic Plan in Ireland” (lines 492-493; p.12).
Similarly as above, the authors should consider providing references for the “Referendum to limit land take in Switzerland” (lines 529; p.13), the “Weber Law in Switzerland” (lines 538-545; p.13), the “Sofia General Urban Development Plan” (line 586; p.14) and the “Spatial Plan of Zone Chalupkova” (line 587; p.14).
References could also be helpful in the “re-creation of Lake Karla in Thessaly” (lines 604-606; p.15), the “Enjoy Waltham Project” (lines 606-609; p.15), the “Piano Periferie 1 and 2” (lines 610-614, p.15) and the “BENE – Berlin Programme” (lines 615-618; p.15).
Lines 624-636; p.15. The authors could also include references for the examples of Projects that are presented in this paragraph.
Line 648; p.15. The authors should change the name of the country “Macedonia” with the officially accepted and according to the UN Protocol “North Macedonia”.
Lines 648-655; p.15. The authors may consider providing references for the Projects that are presented in this paragraph.
Concluding Remarks
In this final section the authors summarize the findings and describe conclusions drawn from their study. These conclusions were design to present recommendations and warnings that can be used when designing new interventions and can be take in account in future developments of land use when aiming for a sustainable urban planning.
As I mentioned in previous sections, there are some instances where the authors present assertions and referring to cases without substantiating them with references. This will help the readers to address to the observed case if they want more details and information.
The authors may wish to provide references for the statements below.
Lines 669-672; p.16: “Densification does not always imply sustainable land use. In some cases, ...in favor of buildings”. The authors should consider proving references as examples.
Lines 706-707; p16. The authors state that “there are a series of examples, through Europe, that show...”. The authors may wish to add references that correspond to this support.
References
Reference [2] and Reference [8] are the same (Solly et al. 2020). One must be removed or change and the reference in the main text should be fixed.
Useful References
Karanikola, P., Tampakis, S., Zafeiriou, E., and Akrivouli, K. (2017) Local people attitudes towards wetland management. The case of Lake Karla in Greece. Journal of Environmental Protection and Ecology, 18 (3): 1268-1276.
Papadopoulos, Y. and Dimitriou, E. (2020) A Large-Scale Nature-Based Solution in Agriculture for Sustainable Water Management: The Lake Karla Case. Sustainability 12 (17): 6761.

Author Response
Answer to reviewer 3
We would like to thank the third peer-reviewer for their helpful comments and suggestions and have revised the paper accordingly. Below, we present more in detail the way in which the various comments and suggestions have been taken into account.
Comments and Suggestions for Authors
Summary
The manuscript summarizes a wide range of interventions and evidences of sustainable urbanization practices that the authors gathered through interviews, surveys and literature from European countries. The authors created a database with the information gathered and a scale to help reader evaluate the outcome of the interventions. The authors focus on the interventions scopes, outcomes and objectives and discuss the success or failure of the implemented approach in each case. Based on their results and the analysis and evaluation of the data and cases, they aim to develop guidance for policy makers. In that way the aim of this study is to be use as a useful guide in future designing of interventions considering urban land change.
Merits
The study focuses in sustainable land use and land change in urban areas. This is an important issue as urban areas are increasing in size and thus the way that these areas are designed is crucial, since they affect not only the environment but also our quality of life. The analyses that the authors use are appropriate for this study. The authors use various approaches to examine the interventions that take place in different European countries, their scopes and outcomes. The authors use online surveys and interviews and also examine the available literature. The literature cited is relevant to the study, but there are several instances, which have been noted, in which the authors don’t use appropriate literature to support their examples and cases. These assertions should be substantiating with references and there being noted in detail.
Critique
Introduction
The Introduction provides a good and general background on the main topic. That gives the reader a quick understanding of the main subject. The objectives of the study are clearly defined in the last paragraph.
Line 49; p.2. Citation [8] and citation [2] are referring to the same study. The authors need to check this and update the reference.
This problem has been solved.
Theoretical Framework and Methodology
The technique used to obtain the results seems appropriate for this study, especially since the main focus is to demonstrate the different approaches of land use and urbanization design among the European countries.
It is not vital to the present study but something that could be interesting would be if the authors could demonstrate any differences in the evaluation of the interventions results obtained using the interview technique in contrast to the results of the literature.
The database was filled in on the basis of desk research and an online survey, to which experts from various organizations responded. Then, the collected information was complemented through desk research for all entries. In this light, the level of information for each intervention is of comparable quality.
However, we do agree on the fact that there may be some differences in the level of detail characterising (i) those interventions directly entered by the project team members as a consequence of their direct knowledge (the team members were fully aware of the project structure and ambition and possessed direct knowledge of the case); (ii) those collected through the literature (the team members were fully aware of the project structure and ambition and possessed but DID NOT include direct knowledge of the case); (iii) those entered by third parties (that were NOT fully aware of the project structure and ambition BUT possessed direct knowledge of the case).
In order to make sure that this would not lead to a misleading sample, quality control on the database was performed in multiple phases by multiple persons, to incrementally ensure that the information included were comparable and of good quality.
Results and Discussion
The Results and Discussion Section is well constructed; divided in subsections using the Scope and goals and Type of Instrument factors as levels of categorization. The authors state in Section 2 that “the interventions were further analyzed by reviewing available online documentation”. However there are some instances that the authors did not substantiate their assertions with references and it is not clear to the reader when discussing their results whether a statement is supported from an interviewee’s survey or from literature.
In relation to this issue and to the more detailed comments and suggestions that follow concerning references: additional references have been added in relation to all interventions for which the success has been assessed on the basis of desk research.
When this is not the case, it means that the proposed argument is based exclusively on the information provided by the experts through the online survey.
We hope the reviewer will consider our effort and the provided solution satisfactory.
Figure 3; p.7. The map is not very easy to read and symbols in neighbor countries in some instances are mixed up and not easy to distinguish between them (in example Belgium and Netherlands).
The comment is valid but the map is the official one from the ESPON SUPER Guide and we are not in the position to change it.
Table 3; p.7. This table presents absolute values. The numbers sum up the interventions that correspond to the degree of success (in a scale from 0 to 5) of the different types of scopes and instruments. But inside the study’s text the authors are using percentages to explain and present the results of this table. Thus is difficult to the reader to correlate the table with the text. The authors could consider adding percentages in the Table’s results.
The table has been modified and the data is now presented in percentages.
Lines 264-266; p.9. The authors state that “this can be seen in the Croatian Spatial Plan...” This statement needs support with references.
As stated above, some new references have been added, as well as two explanatory sentences in Section 4.1.
Lines 311-310; p.9. The authors mention that the “regeneration process in Casoria produced very positive results...” and that “the rehabilitation of parts of the That-el-Kale Quarter in Nicosia... less successful”. This statement needs support with references.
See explanatory comment above.
The same as above applies to the “Case of the 2007 zero-growth plan of Cassinetta di Lugangano (lines 324-325; p.9) and in “the 22@Barcelona regeneration program” (lines 329-330; p.9).
See explanatory comment above.
The authors should also consider adding references to the examples of “the case of Vision Rheintal (Austria)” (lines 352-353; p.10) and “the 2014 Tuscany Regional Law” (lines 358-359; p.10).
See explanatory comment above.
References also must cited that describe the interventions of: “Mini-Holland in Waltham Forest” (lines 422-423, p.11), the “Slovenian Sustainable Urban Mobility Plans” (lines 424-425; p.10), the “City of Sofia’s Underground metro” (lines 431-432; p.10), the “Lyon-Torino high speed railway and tunnel project” (lines 432-435; p.11), the “BOKS-Soil Protection Concept” (lines 437-438; p.11) and the “Lower Austrian spatial planning ordinance” (lines 440-444; p.11), and the “project Green cross-border area”, the “Soil Enhancement Plant of Austria” and the “Tisza River in Hungary” (lines 445-451; p.12),
See explanatory comment above.
Intervention examples that need support with references are also: the “Tri City metropolitan area planning in Poland” (lines 465-466; p.12), the “Corona Verde case in Italy” (lines 474-478; p.12), the “Kooperationsplattform Stadregion in Austria” (lines 478-480; p.12), the “New Finger Plan of Copenhagen” (lines 490-491; p.12) and the “Strategic Plan in Ireland” (lines 492-493; p.12).
See explanatory comment above.
Similarly as above, the authors should consider providing references for the “Referendum to limit land take in Switzerland” (lines 529; p.13), the “Weber Law in Switzerland” (lines 538-545; p.13), the “Sofia General Urban Development Plan” (line 586; p.14) and the “Spatial Plan of Zone Chalupkova” (line 587; p.14).
See explanatory comment above.
References could also be helpful in the “re-creation of Lake Karla in Thessaly” (lines 604-606; p.15), the “Enjoy Waltham Project” (lines 606-609; p.15), the “Piano Periferie 1 and 2” (lines 610-614, p.15) and the “BENE – Berlin Programme” (lines 615-618; p.15).
See explanatory comment above.
Lines 624-636; p.15. The authors could also include references for the examples of Projects that are presented in this paragraph.
See explanatory comment above.
Line 648; p.15. The authors should change the name of the country “Macedonia” with the officially accepted and according to the UN Protocol “North Macedonia”.
The text has been changed accordingly.
Lines 648-655; p.15. The authors may consider providing references for the Projects that are presented in this paragraph.
See explanatory comment above.
Concluding Remarks
In this final section the authors summarize the findings and describe conclusions drawn from their study. These conclusions were design to present recommendations and warnings that can be used when designing new interventions and can be take in account in future developments of land use when aiming for a sustainable urban planning.
As I mentioned in previous sections, there are some instances where the authors present assertions and referring to cases without substantiating them with references. This will help the readers to address to the observed case if they want more details and information.
The conclusions have been further substantiated through the inclusion of additional references from the literature, as well as further references to the interventions analysed in the text.
The authors may wish to provide references for the statements below.
Lines 669-672; p.16: “Densification does not always imply sustainable land use. In some cases, ...in favor of buildings”. The authors should consider proving references as examples.
See explanatory comment above.
Lines 706-707; p16. The authors state that “there are a series of examples, through Europe, that show...”. The authors may wish to add references that correspond to this support.
See explanatory comment above.
References
Reference [2] and Reference [8] are the same (Solly et al. 2020). One must be removed or change and the reference in the main text should be fixed.
One reference has been removed.
Useful References
Karanikola, P., Tampakis, S., Zafeiriou, E., and Akrivouli, K. (2017) Local people attitudes towards wetland management. The case of Lake Karla in Greece. Journal of Environmental Protection and Ecology, 18 (3): 1268-1276.
Papadopoulos, Y. and Dimitriou, E. (2020) A Large-Scale Nature-Based Solution in Agriculture for Sustainable Water Management: The Lake Karla Case. Sustainability 12 (17): 6761.
The suggested references have been used in the text and added to the reference list.
Reviewer 4 Report
The paper entitled “Towards sustainable urbanization. Learning from what’s out there” carefully analyses several examples of sustainable urbanization interventions in order to develop a set of recommendation for decision and policy–makers in the perspective to promote a sustainable use of land.
I think it’s a very interesting study and with current relevance. The article is well structured, understandable and clear.
Maybe, the authors could further expand the study by elaborating a feasibility scale for each intervention in relation to some of the European countries investigated and thus in relation to the different aspects that characterise them. However, the current contribution is already sufficient for an original publication with an interesting contribution.
Finally, only a little comment should be addressed by the authors before publication.
SPECIFIC COMMENTS:
Table 3. Within the table it’s possible to see the numbers of interventions which were more or less successful, right? If so, I think it is clearer to indicate the total number of interventions analysed under each type of scope and goals. Moreover, could you specify what is meant by n.a?
Author Response
Answer to reviewer 4
We would like to thank the fourth peer-reviewer for their helpful comments and suggestions and have revised the paper accordingly. Below, we present more in detail the way in which the various comments and suggestions have been taken into account.
Comments and Suggestions for Authors
The paper entitled “Towards sustainable urbanization. Learning from what’s out there” carefully analyses several examples of sustainable urbanization interventions in order to develop a set of recommendation for decision and policy–makers in the perspective to promote a sustainable use of land.
I think it’s a very interesting study and with current relevance. The article is well structured, understandable and clear.
Maybe, the authors could further expand the study by elaborating a feasibility scale for each intervention in relation to some of the European countries investigated and thus in relation to the different aspects that characterise them. However, the current contribution is already sufficient for an original publication with an interesting contribution.
We thank the reviewer for this interesting suggestion and agree that it is not essential to the present study.
Finally, only a little comment should be addressed by the authors before publication.
SPECIFIC COMMENTS:
Table 3. Within the table it’s possible to see the numbers of interventions which were more or less successful, right? If so, I think it is clearer to indicate the total number of interventions analysed under each type of scope and goals. Moreover, could you specify what is meant by n.a?
The table has been changed and it now presents the percentages instead of the number of interventions under each category.
Round 2
Reviewer 1 Report
This paper present merits for final publication BUT need some additional sentences in the theory section about the concept of SUSTAINABILITY IN THE RECENT EU REGIONAL AND URBAN POLITICS. in this context please explain in the conclusion sections the opporrunity of the paper. One sentence.
Author Response
Answer to round 2 reviewer
We thank the peer-reviewer for the new comments and have revised the paper as follows.
Comments and Suggestions for Authors
This paper present merits for final publication BUT need some additional sentences in the theory section about the concept of SUSTAINABILITY IN THE RECENT EU REGIONAL AND URBAN POLITICS. in this context please explain in the conclusion sections the opporrunity of the paper. One sentence.
Additional sentences and references as regards sustainability in recent EU regional and urban politics (see lines 87-91 and 111-114) have been added to Section 2 Theoretical Framework. A further sentence on the opportuneness of the paper has been added with a reference to Section 5 Concluding Remarks (see lines 844-847), creating a new final paragraph.
On line 146 ‘spects,’ has been deleted and on line 693 ‘neighborhood’ has been changed to ‘neighbourhood’.